# A sandbox for prediction and integration of DNA, RNA, and protein data in single cells

**Malte D. Luecken**[1*]**, Daniel B. Burkhardt**[2*]**, Robrecht Cannoodt**[3,4,5*]**, Christopher Lance**[1*]**,
Aditi Agrawal**[6]**, Hananeh Aliee**[1]**, Ann T. Chen**[6]**, Louise Deconinck**[4,5]**, Angela M. Detweiler**[6]**,
Alejandro Granados**[6]**, Shelly Huynh**[6]**, Laura Isacco**[2]**, Yang Joon Kim**[6,8]**, Dominik Klein**[1]**,
Bony De Kumar**[7]**, Sunil Kuppasani**[2]**, Heiko Lickert**[1]**, Aaron McGeever**[6]**, Honey Mekonen**[6]**,
Joaquin Caceres Melgarejo**[7]**, Maurizio Morri**[6]**, Michaela Mueller**[1]**, Norma F. Neff**[6]**, Sheryl
Paul**[6]**, Bastian Rieck**[9]**, Kaylie Schneider**[2]**, Scott Steelman**[2]**, Michael Sterr**[1]**, Dan J. Treacy**[2]**,
Alexander Tong**[7]**, Alexandra-Chloé Villani**[10]**, Guilin Wang**[7]**, Jia Yan**[6]**, Ce Zhang**[7]**, Angela O.
Pisco**[6†]**, Smita Krishnaswamy**[7†]**, Fabian J. Theis**[1†]**, Jonathan M. Bloom**[2†]

[1]Helmholtz Center Munich, [2]Cellarity, [3]Data Intuitive, [4]VIB Center for Inflammation Research,
[5]Ghent University, [6]CZ Biohub, [7]Yale University, [8]UC Berkeley, [9]ETH Zurich, [10]Harvard Medical
School, [*,†]Equal Contribution,
{malte.luecken,christopher.lance,fabian.theis}@helmholtz-muenchen.de;
{dburkhardt,jbloom}@cellarity.com; robrecht@data-intuitive.com;
angela.pisco@czbiohub.org; smita.krishnaswamy@yale.edu

## Abstract

The last decade has witnessed a technological arms race to encode the molecular
states of cells into DNA libraries, turning DNA sequencers into scalable single-cell
microscopes. Single-cell measurement of chromatin accessibility (DNA), gene
expression (RNA), and proteins has revealed rich cellular diversity across tissues,
organisms, and disease states. However, single-cell data poses a unique set of
challenges. A dataset may comprise millions of cells with tens of thousands of
sparse features. Identifying biologically relevant signals from the background
sources of technical noise requires innovation in predictive and representational
learning. Furthermore, unlike in machine vision or natural language processing,
biological ground truth is limited. Here we leverage recent advances in multi-modal
single-cell technologies which, by simultaneously measuring two layers of cellular
processing in each cell, provide ground truth analogous to language translation.
We define three key tasks to predict one modality from another and learn integrated
representations of cellular state. We also generate a novel dataset of the human
bone marrow specifically designed for benchmarking studies. The dataset and
tasks are accessible through an open-source framework that facilitates centralized
evaluation of community-submitted methods.

## 1 Introduction

Humans reliably develop from a single cell to about 37 trillion cells that collectively manifest
movement, immunity, and thought [1]. The 20th-century development of molecular biology revealed
DNA as the evolving instructions for life, with genes transcribed to RNA that is translated into
proteins. In turn, these proteins perform critical cellular functions. In addition to propagating neural
signals, mediating immune function, or contracting muscle fibers, proteins are regulators of gene
expression. Transcription factor proteins turn genes on and off in response to environmental signals
and in the course of differentiation. Indeed, a fundamental challenge of biology and medicine is to
understand the cellular programs whereby the same DNA source code gives rise to the incredible
diversity of cell types and states.

35th Conference on Neural Information Processing Systems (NeurIPS 2021) Track on Datasets and Benchmarks.

This genetic regulation is among the complex dynamical systems in the universe. A single human cell contains 6.2 billion base pairs of DNA of which 1.2% encodes roughly 25 thousand protein-coding genes with the remaining 98.8% having regulatory or unknown function, if any [2]. In that same cell, there are hundreds of thousands of messenger RNA molecules and hundreds of millions of protein molecules. Dynamic regulation happens at each level in this process [3]. Epigenetic modifications on DNA determine local accessibility to transcription factor binding and RNA transcription. RNA molecules are then further modified to regulate the rate at which the transcripts are translated into proteins. Proteins are also modified to alter their regulatory functions, which include organizing DNA in space, modifying RNA and other proteins, forming complexes (including RNA polymerase), and binding to specific DNA sequences to promote or suppress gene expression.

A decade ago, techniques emerged to encode the molecular states of individual cells into DNA libraries, thereby turning DNA sequencers into single-cell microscopes. These molecular states span multiple modalities: the level of accessibility along the entire genome to regulatory and transcriptional proteins (chromatin state), the number of RNA molecules per gene for all genes, and the number of molecules per protein for hundreds of species of protein. The incredible scaling of single-cell measurement technologies, far exceeding Moore's law, has moved the field from a "small N, large P" into the big data regime [4]. Some datasets measuring one modality now include millions of cells.

The growth of single-cell data has fueled the development of statistical models and algorithms [5]. Yet, many barriers exist for data science at single-cell resolution [6]. Although cells are information dense, their minuscule content leads to measurement error and uncertainty. Furthermore, the readouts are high dimensional, requiring algorithms to scale across both observations and features. Additionally, the noise patterns in single-cell data arise at the level of features, observations, and groups of observations handled in batches. These patterns are not well understood and can have large effects [7], requiring novel methods to disentangle biological variation from technical noise.

As method developers strive to develop innovative methods, molecular biologists continue to push the boundaries on what information can be measured in individual cells. One of the most powerful recent advances in single-cell technologies is simultaneous measurement of multiple modalities in the same cell [8, 9]. The first multi-modal single-cell technology was introduced by [8], jointly profiling RNA gene expression (GEX) and cell surface protein markers using antibody-derived tags (ADT) compatible with high-throughput droplet-based technologies. Newer techniques enable joint profiling of RNA gene expression and genome-wide DNA accessibility (referred to as ATAC: assay for transposase-accessible chromatin) [10, 9]. Measuring multiple layers of the genetic regulatory process simultaneously in single cells offers new opportunities to study the regulatory processes governing life. However, few tools yet exist to fully leverage the potential of multimodal single-cell data.

Here we aim to drive machine learning innovation in this field of molecular and cellular biology using the Common Task Framework (CTF) [11]. In the CTF, a task comprises (1) a public training dataset with ground truth, (2) a private testing dataset, (3) a public challenge in which competitors aim to infer a predictive model from the training data, and (4) a scoring process that quantifies the accuracy of predictions relative to the ground truth. While this framework has been crucial to the success of machine learning innovation in technology and business applications, it has been largely absent in life science, in part due to barriers to assembling, sharing, or even measuring ground truth data at scale (notable exceptions are protein folding [12] and image analysis [13, 14]).

Multi-modal measurement holds promise for molecular biology through a CTF combining aspects of language translation and representation learning. We emphasize three key tasks (**Figure 1**):

1. *Predicting one modality from another.* Accurate predictive models may elucidate principles of genetic regulation and augment the value of existing and future single-modality datasets, which are simpler and cheaper to generate.

2. *Matching cells between modalities.* Inference of the true pairing between modalities of jointly measured cells enables alignment of single-modality datasets for multi-modal analysis.

3. *Jointly learning representations of cellular identity.* Complementary layers of information may be combined to learn more meaningful representations of cellular states and dynamics.

The CTF requires a high-quality benchmark dataset. Multi-site preparation of the dataset is crucial for developing methods that generalize across lab-specific technical noise. The largest multi-omic

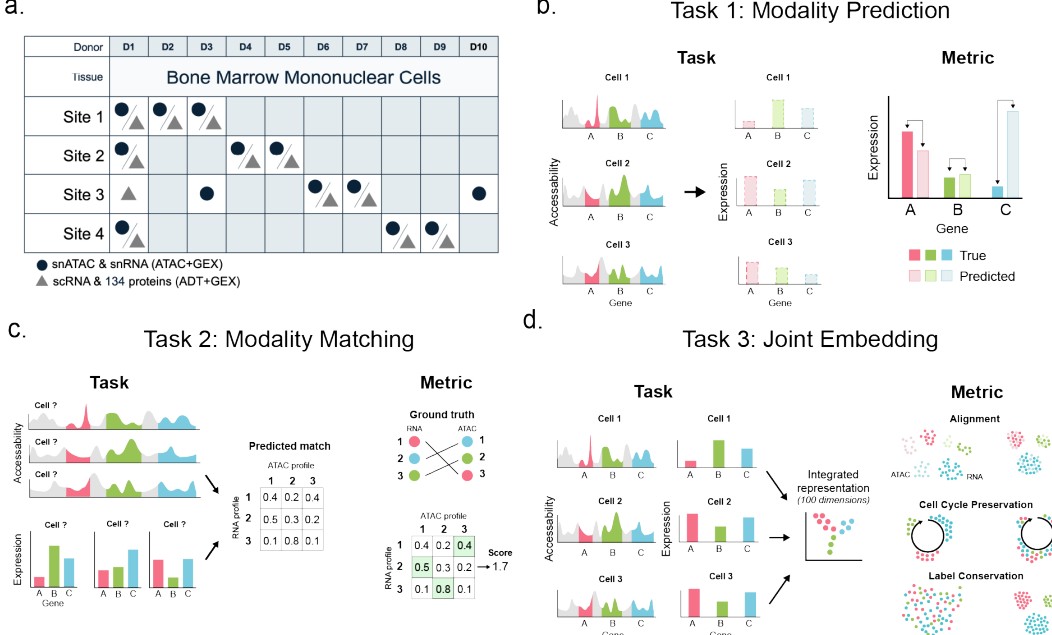

Figure 1: Components of the method development sandbox for single-cell multi-modal data integration. These include **(a)** the first multi-modal BMMC reference dataset with multiple batches and ground-truth annotations, and **(b-d)** three defined multi-modal integration tasks with 19 metrics to evaluate success (not all metrics shown).

dataset (ATAC+GEX) profiles 34,774 cells using a non-commercially available technology measured in a single laboratory [15]. To date, the largest multimodal dataset is 211,000 peripheral blood mononuclear cells (PBMCs) profiled using ADT+GEX by [16], also in a single facility. This reference dataset contains up to 288 protein markers, but PBMCs are a highly differentiated tissue characterized by strong cluster structure. To capture regulatory complexity, it is important to also capture developing cells.

To overcome these limitations, we introduce a first-of-its-kind multimodal benchmark dataset of 120,000 single cells from the human bone marrow of 10 diverse donors measured with two commercially-available multi-modal technologies: nuclear GEX with joint ATAC, and cellular GEX with joint ADT profiles. This dataset is multi-site, has a private test split, and captures both developing and differentiated cell types. Data collection was performed using a standardized protocol and commercially available reagents to facilitate replication studies.

In the following sections, we present a sandbox to advance single-cell science using multi-modal data. We first survey prior work in multi-modal single-cell analysis and benchmarking. We next describe our fit-for-purpose multi-donor, multi-site, multi-modal bone marrow dataset. We further motivate and formalize the three tasks above. Finally, we present an extensible computational framework to support centralized benchmarking of community-submitted single-cell methods. We have combined these data, tasks, and infrastructure into a CTF, the first NeurIPS competition featuring single-cell data. Details on the competition and the dataset, including download instructions can be found at https://openproblems.bio/neurips.

## 2 Prior work

### 2.1 The common task framework in the life sciences

The common task framework has driven machine learning as a field and in a breadth of applications. However, relatively few competitions have focused on biological problems and data; indeed, the only previous such NeurIPS competition was the 2019 machine vision task of matching experimental replicates of high-content images of perturbed cell lines [14]. With the recent success of AlphaFold 2 [12], perhaps the most well-known competition in the life sciences is the Critical Assessment of

protein Structure Prediction (CASP) [17], taking place every two years since 1994. There has also been growing interest in Dialogue on Reverse-Engineering Assessment and Methods (DREAM) Challenges [18] as an alternative to Kaggle for the life sciences. These 88 challenges adhere to the CTF but have mainly focused on pharmacology and electronic health records. More recently, a group described a series of single-cell hackathons with a focus on integrating spatial and RNA measurements and concluded that multi-modal benchmarks in cell biology are lacking and critical [19].

## 2.2 Ground truth in single-cell benchmarks

Benchmarks of single-cell analysis methods typically reside in papers that report on new methods or compare a set of existing methods to guide analysts in tool selection [20]. These studies typically rely on four kinds of "ground truth" data:

1. *Fully simulated data* is free and flexible to test specific hypotheses of method utility (e.g., [21]). However, simulated data is only as useful for discovery as our generative understanding of cell biology, hence of limited value on more complex tasks [7].

2. *Synthetically modified real data* creates ground truth by, for example, simulating changes for differential expression algorithms [22] or dropping out data for imputation algorithms [23]. The data distributions are often realistic, but the experimental effects may be oversimplified.

3. *Real data with low-dimensional ground truth* may be generated, for example, by mixing cells from different species to ensure obvious ground truth or by using barcodes to mark cell lineage. These approaches are used to test experimental protocols [24, 25] and to benchmark methods like batch integration [26], deconvolution [27], and lineage inference [28].

4. *Real data with manually annotated labels* provides the most realistic ground truth. However, scale is limited by bandwidth of experts, and even experts disagree on ground truth. For example, literature-derived marker genes continue to rapidly evolve even in well-studied systems. Inconsistent approaches to annotation make it challenging to harmonize independently published studies (e.g., [29]). Complete re-annotation of independent datasets is labor intensive (e.g., [7]).

Notably, ground truth dynamics of the same cell throughout its lifetime are absent, because all existing genome-wide technologies are destructive to the cells.

Technology enabling joint measurement of adjacent levels of cellular processing in the same cell provides a promising form of high-dimensional ground truth, akin to matched documents in machine translation when predicting one level from the other. The first large-scale benchmark dataset of gene expression measured jointly with 228 protein was recently published [16]. Here, we measure the accessibility of 119,254 genomic regions, the expression of 15,189 genes, and the abundance of 134 surface proteins with ATAC+GEX and ADT+GEX in a multi-site, multi-donor dataset of a complex biological system.

## 2.3 Multi-modal single-cell analysis

Recent multi-modal computational methods were designed to integrate measurements of proteins and RNA to learn joint latent representations of cellular state [30, 31, 32, 16, 33], infer gene regulation [34], and infer unmeasured modalities [35, 16, 33]. Approaches include factor analysis [32, 16, 34] and unsupervised neural network architectures [31, 30, 35, 33] to embed cells measured with each modality into a common space. As long as fit-for-purpose benchmarks are absent, it remains unclear how well these methods handle continuous cellular phenotypes and complex batch effects. Several techniques have been proposed for the analysis of jointly profiled multimodal single-cell data. These methods use neural networks to embed multimodal data into a joint latent space using interoperable encoders and decoders [35] or a VAE [36]. Another recently described approach builds a graph within and across modalities using a weighting based on the information content identified in local neighborhoods in each modality [16].

## 3 Overview of the multi-modal single-cell analysis sandbox

Our work aims to advance multi-modal single-cell data science through the CTF. This requires identifying relevant public datasets, generating a fit-for-purpose dataset that includes privately held test

data, formalizing tasks with biological relevance, and creating a computational framework to support benchmarking of community-contributed methods. The result of this work is a flexible sandbox to support method developers from the machine learning and computational biology communities toward understanding regulatory biology.

## 3.1 Generating a multi-modal single-cell benchmark dataset

The utility of a benchmark dataset is driven by its fidelity to real world tasks [37]. In our context, this means ensuring that our benchmark dataset captures the core complexities of single-cell datasets. Furthermore, raw data must be processed, annotated, and formatted to be usable by machine learning methods. As standards in single-cell analysis are rapidly evolving, we leveraged our previous work identifying best practices [20], convened an expert committee of scientists from Helmholtz, Yale, Chan Zuckerberg Biohub, VIB–Ghent University, and Cellarity, and consulted additional experts from Helmholtz Center Munich, Harvard, the Sanger Institute, and Stanford University to assist with cell annotation. The result of this effort is a high-quality, fit-for-purpose benchmark dataset for multi-modal single-cell analysis.

**Considerations for data generation**    We identified seven categories of desiderata for a multi-modal single-cell benchmark dataset:

1. *Multiple modalities* should capture causally-related layers giving complementary views into cellular processing and state.
2. *Continuous biological processes* are central to the differentiation and functioning of cells and tissues. Relative to clusters of discrete cell types, continuous changes in cellular profiles are easily mistaken for noise. Our dataset should include well-studied continuous processes that we can unambiguously annotate across samples.
3. *Complex batch effects* are a critical challenge in single-cell data analysis [7]. The size of a batch is limited by the device used to generate the data and the capacity of the data generator to process samples concurrently. Thus, especially in multi-lab collaborations, complex, nested batch effects are the norm.
4. *Human donor diversity* in genetic background, age, sex, and lifestyle also impact variability at the single-cell level. Our dataset should represent this variability while controlling for disease and smoking status, mirroring a typical experimental study design.
5. *Disease-relevance* of the biological system raises exciting possibilities for translating biological understanding to improve human health.
6. *Accessible, state-of-the-art protocols* are critical to ensure our dataset remains relevant and extensible, given the pace of technological innovation.
7. *Open access* to the dataset through informed consent ethics statements is essential.

From these criteria, we selected bone marrow mononuclear cells (BMMCs) as our tissue. Bone marrow is the site of several stages of erythrocyte differentiation and B cell maturation, continuous biological processes that are represented in a complementary fashion across modalities: differentiation from a multi-potent progenitor state into a particular developmental lineage (e.g., committing to the erythrocyte lineage from hematopoietic stem cells) requires large-scale chromatin remodeling (measured by ATAC). Additionally, protein measurements are known to improve the representation of immune cell states over transcription alone [16]. Bone marrow is the site of multiple diseases, including leukemia (cancer leading to abnormal white blood cells), myeloproliferative disorders (too many white blood cells), and aplastic anemia (lack of red blood cells). Improved representations of immune cell development may also aid the modeling of complex immune responses to diseases such as COVID-19. Moreover, BMMCs may be ethically sourced from commercial vendors, such that single-cell data with anonymized metadata can be freely shared.

We sourced multiple samples of BMMCs from 10 donors via AllCells (California, USA), all healthy non-smokers without recent medical treatment. Donors varied by age (22 - 40), sex, and ethnicity (details in the associated datasheet). For each sample, we generated joint ATAC+GEX and ADT+GEX measurements, thereby producing paired sets of joint multi-omic data from each donor.

Each experiment was loaded to target a recovery of 7,000 cells per measurement and sample, leading to a target dataset size of 150,000 multi-modal cellular profiles. Preprocessing removes, on average,

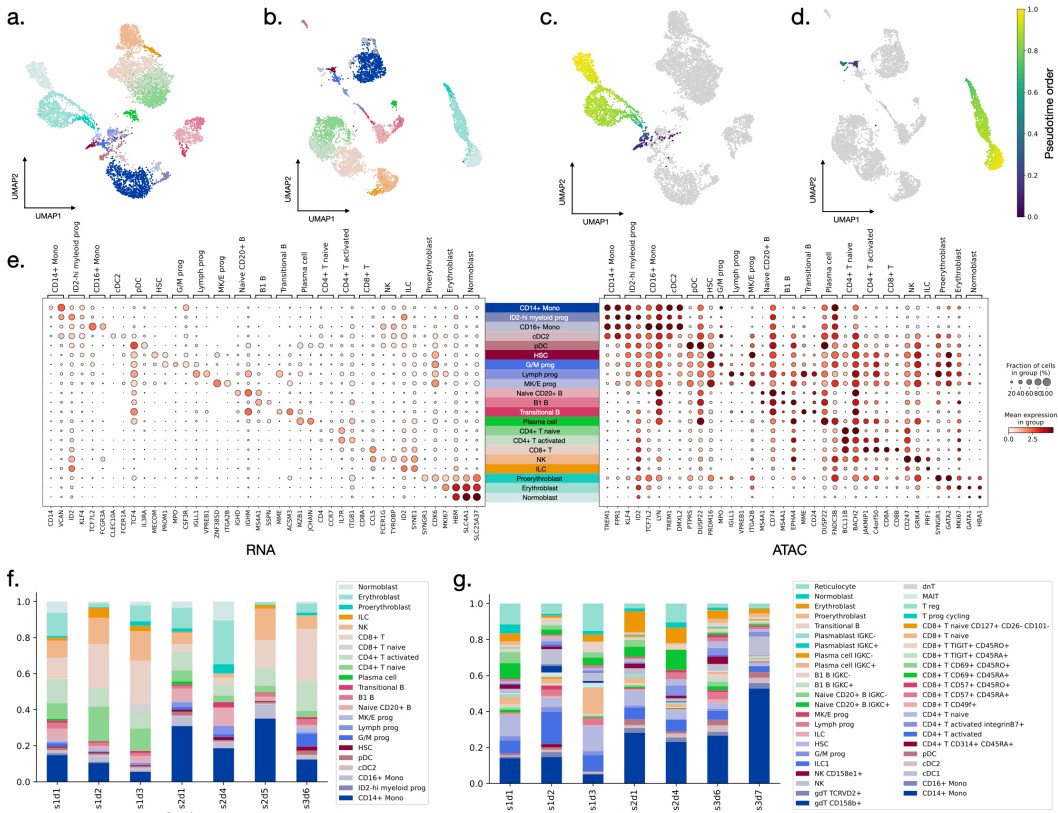

Figure 2: Annotation of ground truth cell types, states, and trajectories in 10X Multiome data from Site 1, Donor 1. Two dimensional UMAP representations of the (**a**). RNA and (**b**) ATAC data show the cell types annotated for this donor. Cellular identity was further quantified by position along the erythrocyte development lineage for a subset of cells as shown for (**c**) RNA and (**d**) ATAC on a UMAP embedding coloured by the pseudotime ordering of cells indicating progress along this trajectory. The literature and data-derived cell identity markers shown in the dotplot in (**e**) were used to perform the cellular identity annotation. Cell identity composition of (**f**) 6 10X multiome samples and (**g**) 2 CITE-seq samples. Abbreviations: B - B cell; T - T cell; Mono - Monocyte; prog - progenitor; HSC - Hematopoietic stem cell; HSPC - Hematopoietic stem and progenitor cell; ILC - Innate lymphoid cell; Lymph - Lymphoid; MK/E - Megakaryocyte and Erythrocyte; NK - Natural Killer cell; cDC2 - Classical dendritic cell type 2; pDCs - Plasmacytoid dencritic cells.

15-30% of the putative cell profiles, leading to an estimated final dataset size of 120,000. This number will be updated when the dataset processing is completed.

Detailed experimental protocols may be found in the Supplementary Materials and will be deposited at the public protocol sharing platform `protocols.io` shortly after submission. Finally, we introduced nested batch effects into our experimental design by generating 3 samples of data each at 3 different sites in the US and 1 in Germany (**Figure 1**). Samples from one of the donors were measured at all sites to capture site-specific batch variation, while each site measured three distinct donors to capture within-site donor variation. To our knowledge, this BMMC benchmark dataset is the most comprehensive multi-modal benchmark dataset ever generated.

**Processing, annotation, and splits of the benchmark dataset**    Raw chromatin accessibility, gene expression, and protein abundance data were processed and analyzed using our previously published best practices [20] and a pipeline set up from the Scanpy and Signac platforms [38, 39] as a basis for quality control, normalization, dimensionality reduction, clustering, feature selection, and trajectory inference.

We generated ground-truth cell identity labels by annotating cellular types and states (**Figure 2a,b**) via state-of-the-art analysis pipelines using literature, data-derived, and expert curated marker genes,

and annotating the erythrocyte development trajectory (**Figure 2c,d**). For benchmarking the third (joint representation) task, it is crucial that ground-truth biological annotations are generated for each batch and modality separately, relying on a feature-based definition of cellular identity derived from the literature and our data (**Figure 2e**). Although time-intensive–roughly 4 days per dataset for a PhD student analyst–this avoids relying on a joint representation method for annotation, which is the standard in the field. A full description of the analysis can be found in **Section A.1**. All analysis pipelines are provided as reproducible Jupyter notebooks at `https://github.com/openproblems-bio/neurips2021-notebooks`.

Each sample contains broadly the same cellular identities in varying proportions (**Figure 2f,g**). Profiles of cells with the same identity within a sample exhibit stochastic biological and measurement variability. Across samples, differences are also driven by batch effects. The distribution of samples across donor and data generation sites (**Figure 1**) facilitates train-test splits of increasing difficulty to model and evaluate critical forms of real-world generalization: within sample, within site across donor, within donor across site, and across donor and site.

**Challenges with generating a benchmark dataset** Generating a multi-modal single-cell benchmark dataset poses a unique set of challenges. Sourcing reagents involves working with multiple commercial vendors with a supply chain impacted by the COVID-19 pandemic and a $< -80$ºC cold chain. Generating a sequencing dataset from a human tissue sample is labor intensive, taking roughly three weeks and involving at least three trained scientists to go from tissue to sequencing data ready for computational processing. Preprocessing and annotation take roughly three weeks for first samples and two days for further samples which also require expert guidance and review of biological annotation. Particularly when piloting new technologies, single-cell experiments often fail for reasons that may occur anywhere from sample preparation to sequencing. Finally, these experiments are expensive. Between reagents and labor, this dataset required more than $200,000 in financial support for which we are grateful to the Chan Zuckerberg Initiative, Cellarity, and the participating non-profit institutions. More details and these challenges can be found in **Section A.4**. Nevertheless, we hope others are interested to extend and validate this dataset. We provide recommendations for getting involved in the accompanying datasheet.

## 3.2 Formalizing benchmark tasks and metrics

While many grand challenges in single-cell data science have been articulated [6], the CTF requires mathematically precise definitions of tasks and metrics to drive algorithm development. We now further motivate and formalize our three key multi-modal tasks and related metrics.

**Task 1: Predicting one modality from another** Generally, genetic information flows from DNA to RNA to proteins. DNA must be accessible (ATAC data) to produce RNA (GEX data), and RNA in turn is used as a template to produce protein (ADT data). These processes are regulated by feedback: for example, a protein may bind DNA to prevent the production of more RNA. Methods capable of accurately predicting one modality from another may validate or learn rules governing these complex regulatory processes. Furthermore, such methods may augment the value of existing and future single-modality datasets, which can be generated at high-quality more simply and cheaply.

Formally, the task is to predict all features of one modality based on all features of the second modality. As metrics, we consider root mean squared error (RMSE) and Pearson correlation on log-scaled counts, as well as Spearman correlation.

**Task 2: Matching cells between modalities** Nearly all existing single-cell datasets are single modality, and indeed communities have formed to specifically model chromatin, RNA, or protein data. Aligning observations of different cells with the same identity across modalities would open up paired single-modality datasets to multi-modal data analysis methods leveraging complementary layers of information. This task is further distinguished from modality prediction because not all features are equally relevant for matching cell identities. Understanding how feature selection influences matching accuracy may shed light on the significance of different regions of DNA or transcripts of RNA in cell identity and regulation of downstream genetic processes.

Formally, in the matching task, we present the jointly profiled cells as two sets of unmatched singly profiled cells. The algorithmic goal is assign to each cell in modality one a probability distribution

across all cells in modality two, so as to place high probability on the true matched cell. Hence with $n$ cells, the output format is an $(n, n)$ matrix of non-negative values where each row sums to 1. To manage memory requirements, we enforce sparsity of the matrix to at most 1000 non-zero values per row. As metrics, we consider area under the precision recall curve (AUPR) and the average probability assigned to the correct matching. The latter is a relative measure per dataset that accounts for non-identifiability among cells with the same identity.

**Task 3: Jointly learning representations of cellular identity** Multi-modal measurement holds promise for combining complementary layers of molecular information to learn highly resolved descriptions of the underlying biological states of cells and their collective roles in tissue function. To transfer learning across datasets, encoders must account for and remove batch effects.

Formally, the task is to embed cells into a latent space of 100 dimensions based on all features of two modalities. However, there is no canonical way to measure the quality of a joint embedding. In our previous work, we concluded that a good strategy is to combine metrics of biological conservation and batch correction. Biological conservation metrics quantify how well an embedding captures expertly annotated biology as described in **Section A.1**. We defined five such metrics that assess preservation of annotated cell types, cell cycles, and inferred trajectories in the dataset. Batch correction metrics assess the removal of batch effects in the embedding. A full description of all metrics is in **Section A.1.6**. In the competition, embedding algorithms will be scored as a weighted sum of these metrics as described in **Section A.2.4**.

### 3.2.1 Baseline performance

To provide a baseline for performance in each task, we implemented Positive Controls (PC), which use the ground-truth solutions in order to return (near) perfect predictions, Negative Controls (NC), which return constant or random values to return exceptionally bad predictions, and four Baseline (B) methods, which are a combination of well-established off-the-shelf algorithms (**Figure 3**, appendix). These baseline results provide an upper and lower bound for performance as well framing the relative difficulty of each task and subtask.

### 3.3 Computational framework for centralized benchmarking

Several strategies were used to make the components in this pipeline as robust, reusable and reproducible as possible. 1) We predefined a set of 'component types' and the format of the input/output files that each component expects (**Figure 5**a). 2) Each input/output file is an AnnData [38] file that is required to contain certain fields depending on the component type. 3) Each component is a Viash [40] component which allow for developing components as standalone scripts (e.g. Python, R, Bash) that plug into Nextflow pipelines by using Viash to export them to Nextflow modules (**Figure 5**b). 4) Thanks to the combination of technologies used, the pipeline used to generate the pilot results are exactly the same as is used when evaluation a submission to the competition framework.

A full description of the pipeline may be found in **Section A.3**. Documentation of the components is available on the competition website and accompanying GitHub repository.

### 3.4 Tools to facilitate data access and exploration

During the competition, training splits will be made available via a public Amazon Simple Storage Service (S3) bucket. Download instructions may be found at `https://openproblems.bio/benchmark_dataset`. Each dataset is stored in two AnnData objects [38], one for each modality. After the competition, datasets will be made available at the CZI cellxgene portal at `https://cellxgene.cziscience.com/`.

We have also secured support from Saturn Cloud (New York, NY) to host Jupyter servers preloaded with notebooks for data exploration and analysis. Interested users may go to `https://openproblems.bio/neurips` to find information about how to sign up for a free Saturn Cloud account to access the servers and notebooks.

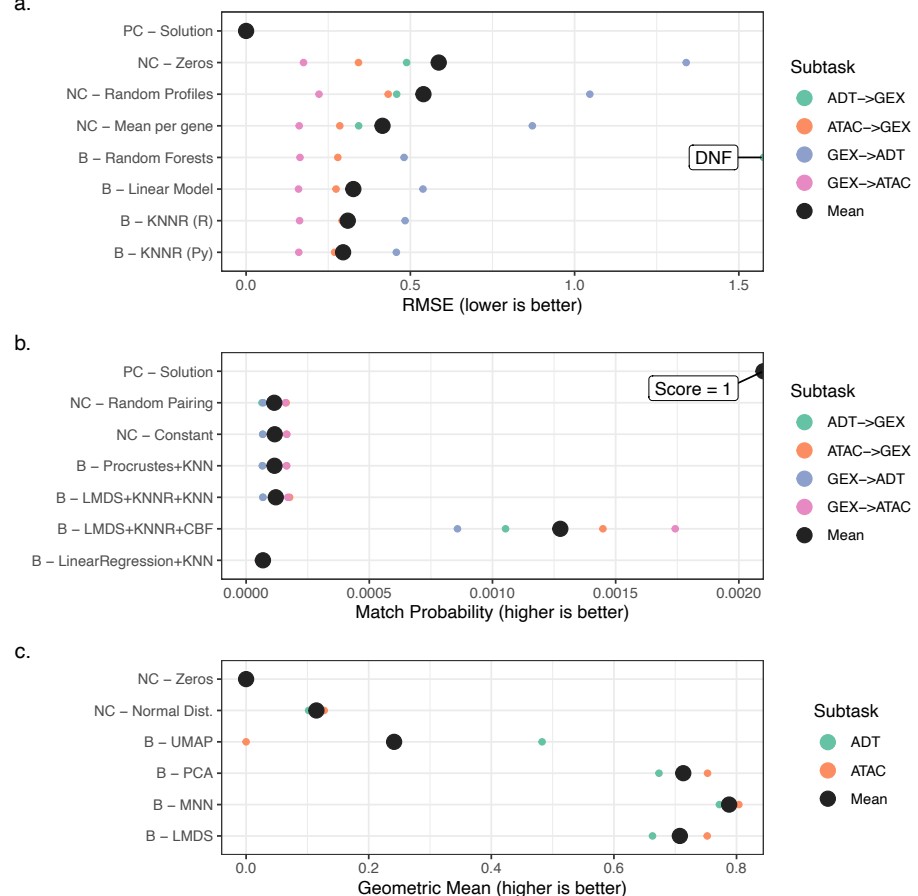

Figure 3: A pilot study on several baseline methods shows that the overall benchmarking pipeline seems to behave as expected; positive controls (PC) perform better than baseline (B) methods and baseline methods perform better than negative controls (NC). **(a)** The pilot results of the Predict Modality task. **(b)** The pilot results of the Match Modality task. **(c)** The pilot results of the Joint Embedding task. The used metric is the geometric mean of the metrics as defined in section 3.2.

## 4  Conclusion

Gene regulation is implemented by high-dimensional dynamical processes that drive the diverse biological functions required for life. Access to measurements of multiple layers of molecular information in single cells is a crucial step toward developing an integrated model of cellular functions. However, this new class of data requires new innovative methods to uncover novel biology. A fundamental challenge in algorithm development is assessing model performance, especially in a cases where ground truth difficult to obtain. Here, we use both the multi-modal nature of jointly profiled cells and expert annotation of a well-studied system to develop a sandbox and NeurIPS competition with three key tasks of multi-modal data integration.

To support these efforts, we generated the largest multi-modal benchmarking dataset currently available with ground truth annotations. This dataset is distinguished by the number of modalities measured, the large number of cells, and the nested batch structure of the study design. This design enables benchmarking of real-world generalization, unprecedented in multi-modal single-cell analysis.

While we have focused on opportunities for machine learning to advance our understanding of biology through the Common Task Framework, we hope access to these fundamental scientific challenges and unique data will also inspire creative new directions for machine learning itself.

# 5    Acknowledgements

We would like to thank Carlos Talavera-Lopez from Helmholtz Munich, Marcela Alcantara from Stanford University, and Rasa Elmentaite from the Sanger Institute, Cambridge, UK for help with interpreting and annotating our BMMC data. Furthermore, we thank Thomas Walzthoeni for support with up-scaling the analysis provided at the Bioinformatics Core Facility, Institute of Computational Biology, Helmholtz Zentrum München and the joint research school Munich School for Data Science (MUDS) supporting CL. This project has been made possible in part by grant number 2021- 235155 from the Chan Zuckerberg Initiative DAF, an advised fund of Silicon Valley Community Foundation and by the Helmholtz Association's Initiative and Networking Fund through Helmholtz AI [ZT-I-PF-5-01] and sparse2big [ZT-I-0007].

# 6    Author Contributions

MDL, DBB, RC, CL, and JMB wrote the paper. AA, BDK, SS, GW, CZ, SH, LI, SK, JCM, KS, DJT, JY, MS, MaM, SS and HL generated the data. MDL, DBB, RC, CL, HA, AC, AG, YJK, AM, BR, and AT analysed the data under supervision of MDL, FJT, AOP, and ACV. RC, DBB, LD, CL, AG, MiM, BR, MDL, and AT built the infrastructure and ran the pilot study. DBB, MDL, SK, JMB, FJT, and AOP coordinated the project. All authors read and reviewed the final manuscript.

# 7    Competing Interests

FJT reports receiving consulting fees from ImmunAI and ownership interest in Dermagnostix GmbH and Cellarity. DBB, LI, SK, KS, SS, DJT, and JMB report being employed by and having ownership interest in Cellarity.

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
