# OpenReview forum: "A sandbox for prediction and integration of DNA, RNA, and proteins in single cells"
_NeurIPS.cc/2021/Track/Datasets_and_Benchmarks/Round2 — NeurIPS 2021 Datasets and Benchmarks Track (Round 2)_

### Official Review · Reviewer_tw6R · 2021-09-20
**Updated Review of "A sandbox for prediction and integration of DNA, RNA, and protein data in single cells"**

**Rating:** 8
**Confidence:** 3

**Strengths:**

The dataset is large, original, and was collected using the latest technology to profile DNA accessibility, RNA and protein expression from single cells. The adherence to the Common Task Framework is another strength with their clearly-defined tasks and evaluation metrics. The dataset was rigorously constructed using multiple institutions, multiple donors, expert annotation/processing, and an experimental design that nested batch effects to capture the many levels of biological and technical variation inherent in single-cell data. In addition, the free compute from Saturn Cloud will hopefully stimulate global participation and bright minds from all countries and socioeconomic groups. Overall, the data curation, processing, and expert-annotation represent a tour de force of molecular biology, single-cell technology, and computational genomics.

**Weaknesses:**

There is very little description of initial performance benchmarks for the proposed tasks, however I was able to find the "dummy baselines" on the Jupyter notebooks hosted on GitHub. As this is primarily seen to be a dataset contribution, state-of-the-art performance is not expected, but including some baseline metrics for the tasks could help readers unfamiliar with the field appreciate the difficulty of problem.

Ethical and societal issues as well as dataset limitations are discussed, but primarily within the supplemental Dataset Datasheet. The potential reidentification of research participants using genomic data in combination with public genealogical databases or public records is briefly discussed but the ethical and societal implications are not further discussed. Although the researchers will have no direct control over how the data is used once released to public scientific databases, it is recommended to discourage use that may harm the employability, insurability, reputation or financial standing of the research participants (it is common sense, but unfortunately needs to be explicit due to rare bad actors).

Also, more discussion of the dataset limitations would be appreciated. The challenges described in the text begin this discussion, but predominantly focus on the time, labor, and financial cost. What other lessons were learned during creating the benchmark? If the process had to be repeated, what would the authors do differently? How might future creators of benchmark datasets learn from and improve upon this example?

The introduction, though very well-written, could be tightened to allow for any of these topics above to be expanded upon.

**Additional Feedback:**

All feedback is described above.

**Clarity:**

The manuscript was well-written with a detailed description of the prior work and dataset curation as well as the specific tasks involved in the challenge. The manuscript formatting adheres to the guidelines. No spelling or grammar errors were identified.

**Correctness:**

The dataset is rigorously constructed applying the current best practices in the field. The design of the dataset collection with nested batch effects was well thought out. The benchmark tasks and evaluation metrics are well defined in the main text or appendix. The dataset is accessible on the associated website (https://openproblems.bio/neurips).

**Documentation:**

The data curation, processing, and organization are well-documented and described in the Appendix, supplemental Dataset Datasheet, or the challenge website (https://openproblems.bio/neurips). The described dataset hosting, licensing, and maintenance plans are acceptable. The computational benchmarking framework is also nicely described in the Appendix.

**Ethics:**

The ethical issues relating to the dataset are primarily discussed in the supplemental Dataset Datasheet. The supplemental Dataset Datasheet section VII part L states that the research "does not pose a 'high risk' to the rights and freedoms of the donors as defined by the GDPR Guidelines". Unfortunately the GDPR is not explicitly clear but does include the following examples quoted below (bold emphasis added). A DIPA is not required but left to their discretion. Also, as the authors note, appropriate research consent was obtained with a discussion of potential risks.
> If you’re processing personal data related to “racial or ethnic origin, political opinions, religious or philosophical beliefs, or trade union membership, and the **processing of genetic data**, biometric data for the purpose of uniquely identifying a natural person, data concerning health or data concerning a natural person’s sex life or sexual orientation”

There is no clear disclosure of author financial or other competing interests, but the reviewer acknowledges such interests do not detract from the dataset utility or validity.

**Relation To Prior Work:**

The relevant prior work is discussed. This benchmark dataset represents a significant contribution with the potential to stimulate multiple lines of research.

**Summary And Contributions:**

Single-cell technologies have been successfully applied to basic and translational research questions and revealing the complex inner workings of the cell. Multi-modal single-cell technologies are emerging as effective tools for quantifying complex subcellular networks in health and disease. However, there are few large, standardized, multi-modal single cell benchmarks to stimulate algorithm and method development.

Lücken, Burkhardt, Cannoodt, Lance, et al. present a large, multi-institutional, multi-modal dataset profiling single cells from donor bone marrow samples. Among the many strengths of this contribution include the large multi-institutional multi-modal dataset collected using the best practices in the field. The tasks are well defined with clear evaluation metrics according to the Common Task Framework. The large, curated, and well documented multi-modal dataset represents an outstanding contribution to the field.

Updated review comments:
The authors have addressed many of the concerns raised by the reviewers and the revised manuscript is improved. I had similar concerns about specimen number and diversity, however I am familiar with how laborious and expensive it is to acquire and process the data, especially given the structure of the experiments. The baselines are sufficient to set the minimum performance and upper/lower bounds on performance since this is primarily a dataset contribution. I appreciate the thoughtful consideration of ethical issues and the additional internal review described by the authors addresses my concerns. The additional discussion of lessons learned from this benchmark will hopefully help others in the field. As described above, the dataset is large for single cell studies and well-curated with a specific task for benchmarking multi-modal inference in single-cell data. This will surely be a very interesting and useful dataset for multi-modal single-cell gene regulation.

---

> ### Author Response · Authors · 2021-09-29
> **Response to Reviewer tw6R**
>
> We appreciate the reviewer’s enthusiastic comments. We took great care to structure the study design, formulate the tasks, carry out the data analysis, and provide accessible computational resources. We hope this work provides a foundation to drive forward the fields of molecular biology, single-cell analysis, and genomics.
>
> **Re: Performance of baseline methods**
>
> We agree that the initial submission lacked descriptions of baseline performance for each task as we focused primarily on the discussion of the dataset. In response to the reviewer’s concern, have added a section in the appendix describing the performance of 4 baseline methods per task and the performance of dummy methods (e.g. reporting random expression values for task 1). We hope this section provides additional context for the difficulty of the each task.
>
> **Re: Ethical considerations associated with release of human sequencing data**
>
> We are carefully considering how to release the human sequencing data to the public. Currently, only anonymized donor IDs with UMI counts matrices for each modality are publicly available. It is not possible to re-identify any participant based on this data as it contains no genetic markers. We are in discussion with the donation center and CZI to determine whether the data ought to be deposited in dbGaP or EGA, which restrict access to the genomic information. Meanwhile, we emphasize that these donors explicitly consented to submission of the data into an unrestricted access repository. We will also take care to provide guidelines for use alongside the sequencing data that address these privacy concerns.
>
> **Re: Discussion of learnings from the data generation process**
>
> We appreciate the encouragement to share our learnings of this process to the broader research community. We have added a section in the appendix describing these challenges and limitations along with what we would do differently now that we’ve gone through the process once.
>
> **Re: GDPR and DIPA**
>
> We agree with the reviewer that the GDPR is not explicitly clear on the requirement of a DIPA for a research study of this kind. We agree that the data is considered sensitive, therefore we are conducting an additional internal review prior to public disclosure of the genetic data as discussed above.
>
> **Re: Financial or competing interests**
>
> We have added a conflict of interest section.

---

### Official Review · Reviewer_RB4f · 2021-09-21

**Rating:** 6
**Confidence:** 2
**Correctness:** The dataset is generally constructed …
**Clarity:** The paper is well-written and organized.

**Strengths:**

Additional modalities, multi-site data generation, and focus on batch effects are strengths of this dataset over previous datasets.

This dataset should be interesting to researchers working on multi-modal models and computational biology.

Accessibility is a major strength of this benchmark dataset. The use of the Common Task Framework should facilitate the development of machine learning methods for this task, and  centralized evaluation should allow easy and fair comparisons between competing methods as well as provide benchmarks for future work after the conclusion of the challenge.

The social implications is also a strength of this dataset, as advances in understanding cellular transcription and translation are important and may be beneficial to the study of disease, and this dataset may aid in this.


**Weaknesses:**

One concern I have is the diversity of the dataset. It’s not clear to me that cell samples from only 9 donors is enough to create a representative sample.

Another concern I have is that since the dataset only includes data from healthy individuals, this seems to limit the relevance of methods trained on this dataset to the study of disease.






**Additional Feedback:**

There is a typo on 57: “refered” should be “referred”.



**Documentation:**

It is clear how the data is collected and organized. Additionally, instructions for downloading the dataset are provided on their website, with a URL in the paper. There is a plan for hosting and maintenance in the datasheet, as well as information about licensing.

**Ethics:**

I don’t think there are any ethical concerns that warrant further review. The main ethical concern of donor privacy is addressed in the paper.

**Relation To Prior Work:**

It is clear that the proposed dataset differs from existing datasets in that it was collected at multiple sites, has a private test split, and includes both developing and differentiated cell types. It also has three data modalities, as opposed to previous work at that scale which only included two modalities.

**Summary And Contributions:**

The paper proposes a new multimodal dataset of single-cell molecular data that includes RNA gene expression, cell surface protein markers, and DNA accessibility data. The data was obtained from the bone marrow mononuclear cells (BMMCs) of 9 donors and the data was gathered using two different commercially-available technologies (ATAC+GEX, ADT+GEX) at multiple laboratories. Specifically, the accessibility of 119254 genomic regions, the expression of 15189 genes, and the abundance of 134 surface proteins is measured.

The dataset is formatted according to the Common Task Framework (CTF), which has been a standard dataset format for previous datasets such as the Critical Assessment of protein Structure Prediction (CASP) dataset. The proposed tasks are predicting one modality from another, matching cells based on their profiles in different modalities, and creating a representation of cells that captures multiple modalities, along with evaluation metrics for each task. The usage of the dataset is supported by an open-source framework that facilitates the centralized evaluation of methods.

---

> ### Author Response · Authors · 2021-09-29
> **Response to Reviewer RB4f**
>
> We appreciate the reviewers' positive feedback. We agree that the nested batch structure with multiple sites and batches is a major strength. We also appreciated the recognition that by adopting the common task framework and centralized benchmarking framework will provide a long-lasting platform for comparing single-cell analysis methods. Finally, we are similarly enthusiastic about the social impact of applying machine learning to single-cell data. Better understanding the underpinnings of biology is both a fundamental scientific question and has major implications for how we address disease.
>
> **Re: Diversity of the dataset in terms of donor number and health status**
>
> Like the reviewer, we are concerned about the donor diversity within the dataset. First, we would like to note that in terms of cellular diversity, this dataset is one of the largest datasets of its kind. The initial data release of the bone marrow Human Cell Atlas single-cell RNA-seq dataset comprised just over 100K cells from 8 donors measured using a single-modality (PMID: 30243574). We present over 140,000 cells from 9 donors spanning four measurement modalities (ADT, ATAC, cytosolic RNA, and nuclear RNA). Given that multimodal data integration algorithms are trained at the level of individual cells, we believe this dataset well represents the cellular diversity present in the human bone marrow.
>
> Regarding donor variation, we note this dataset was not designed to deeply characterize variation between healthy donors or between healthy or disease samples. Indeed, due to the cost of generating these datasets, the single-cell field has so far focused on describing cellular rather than sample variation, with the samples playing the role of replicates. For example, published multiome datasets typically range from a single donor (DOI: 10.1016/J.CELL.2020.09.056) to 12 mouse samples (DOI: 10.1038/s41587-019-0290-0), while the largest CITE-seq datasets that were recently used to evaluate proposed integration methods incorporate cells from 2 (DOI: 10.1038/s41592-020-01050-x) and 8 donors (DOI: 10.1016/j.cell.2021.04.048). To accurately describe multi-modal differences in the bone marrow between humans, it is critical to first build a high quality reference dataset that captures both cellular diversity and technical noise. Understanding these two axes of variation, we can then understand to what extent variation between different classes of donors represents relevant biology. We are similarly curious about the degree to which these samples vary in response to perturbations, like small molecules or genetic knockouts. We leave these important questions to future work.
>
>
> **Re: Typo on 57**
>
> We appreciate the reviewer’s careful reading and have corrected the typo.

---

### Official Review · Reviewer_iWp6 · 2021-09-21
**An impressive multi-modal dataset for single cell biology**

**Rating:** 7
**Confidence:** 2
**Clarity:** The paper is clear and well written

**Strengths:**

* The authors present a large multi modal bone marrow benchmark dataset
* The authors formulate a collection of novel tasks enabled by the collection of these modalilities with relevant metrics.
* The authors cater for batch effects in the experimental measurements.

**Weaknesses:**

* Some of the biological details in the introduction can be tightened up though it is overall a good treatment of the background suitable for an ML audience.
* I'm hesitant to comment more on the soundness of the paper. Perhaps other reviewers more familiar with the subject can be more probing.

**Additional Feedback:**

This appears to be a large initiative. It would be of interest to the field for the authors to describe in more detail the unique challenges and logistics behind co-ordinating such an effort. For example, they allude to the requirement for the maintenance of a cold chain for some the samples/reagents/

**Correctness:**

* The link https://openproblems.bio/benchmark_dataset is broken
* The paper seems methodologically sound.

**Documentation:**

The dataset appears well-documented.

**Ethics:**

No ethical concerns.

**Relation To Prior Work:**

Previous work is discussed, though I would contend blind evaluation competitions are prevalent in computational biology beyond the examples mentioned.

**Summary And Contributions:**

The authors collect a multi-modal single cell dataset and formulate three tasks (modality prediction, matching and joint embeddings) using these data. The dataset is large and the collection is well-principled. This looks to be an excellent resource for computational biologists as it captures multiple 'layers' of molecular regulation at single-cell resolution.

---

> ### Author Response · Authors · 2021-09-29
> **Response to Reviewer iWp6**
>
> We thank the reviewer for their positive feedback. We also hope that this dataset will serve as a resource to guide computational biologists developing methods to understand multimodal mechanisms of gene regulation.
>
> **Re: Biological description in the Introduction**
>
> We have reviewed the introduction and edited the text for clarity and added citations. We hope the revised text provides a better introduction of the content for an ML audience.
>
> **Re: Broken link**
>
> This link is now fixed.
>
> **Re: Prevalence of blind evaluation competitions in computational biology**
>
> We agree with the reviewer that our original description discounted the prevalence of machine learning benchmarks in computational biology, and we have softened the language used in this section.
>
> However, we still contend that this computational biology makes up a minority of machine learning competitions and with a strong bias towards image classification tasks. We surveyed 100 random Kaggle competitions and found that 14 involved chemistry and life sciences, and 8 of these were image classification tasks where the subject of the images was life science focused. This leaves 6 competitions, two of which involved classification forest cover from cartographic features, two of which focused on brain scans, one focused on classification of audio signals, and one focused on molecular translation which is a computational chemistry task but relevant for drug discovery. The list of competitions surveyed can be found here: https://docs.google.com/spreadsheets/d/1fPPAybeExyktgvjcxxsFge8Nu_bFlufP1lKx2MpLglw/edit?usp=sharing. Based on this survey we believe it’s reasonable to suggest that CTF tasks are not common in the life sciences. If the reviewer can point us to additional blind evaluation tasks, we will gladly include them.
>
> **Re: Challenges and logistics involved with organizing the effort**
>
> We appreciate the reviewers' interest. This was a large initiative requiring deep technical expertise across disciplines. Work was performed across sites with no formal affiliation. We have added a section to the appendix expanding on our key learnings that we will move to the main text in the camera-ready if accepted.

---

### Decision · Program_Chairs · 2021-10-10

**Decision:**

Accept

**Comment:**

There is uniform consensus among reviewers to accept the paper.